# Gender-Dependent Deregulation of Linear and Circular RNA Variants of HOMER1 in the Entorhinal Cortex of Alzheimer’s Disease

**DOI:** 10.3390/ijms22179205

**Published:** 2021-08-26

**Authors:** Amaya Urdánoz-Casado, Javier Sánchez-Ruiz de Gordoa, Maitane Robles, Blanca Acha, Miren Roldan, María Victoria Zelaya, Idoia Blanco-Luquin, Maite Mendioroz

**Affiliations:** 1Neuroepigenetics Laboratory-Navarrabiomed, Complejo Hospitalario de Navarra-IdiSNA (Navarra Institute for Health Research), Universidad Pública de Navarra (UPNA), Pamplona, 31008 Navarra, Spain; amaya.urdanoz.casado@navarra.es (A.U.-C.); jsruizdegordoa@gmail.com (J.S.-R.d.G.); maitane.robles.solano@navarra.es (M.R.); blancacha02@hotmail.com (B.A.); mroldana@navarra.es (M.R.); iblancol@navarra.es (I.B.-L.); 2Department of Neurology, Complejo Hospitalario de Navarra-IdiSNA (Navarra Institute for Health Research), Pamplona, 31008 Navarra, Spain; 3Department of Pathology, Complejo Hospitalario de Navarra-IdiSNA (Navarra Institute for Health Research), Pamplona, 31008 Navarra, Spain; mv.zelaya.huerta@navarra.es

**Keywords:** Alzheimer’s disease, entorhinal cortex, *HOMER1*, sleep deprivation, circRNA, mRNA, calcium, synapses, Amyloid beta, gender, female

## Abstract

The *HOMER1* gene is involved in synaptic plasticity, learning and memory. Recent studies show that circular RNA derived from *HOMER1* (*circHOMER1*) expression is altered in some Alzheimer’s disease (AD) brain regions. In addition, *HOMER1* messenger (mRNA) levels have been associated with β-Amyloid (Aβ) deposits in brain cortical regions. Our aim was to measure the expression levels of *HOMER1* circRNAs and their linear forms in the human AD entorhinal cortex. First, we showed downregulation of *HOMER1B/C* and *HOMER1A* mRNA and hsa_circ_0006916 and hsa_circ_0073127 levels in AD female cases compared to controls by RT-qPCR. A positive correlation was observed between *HOMER1B/C*, *HOMER1A* mRNA, and hsa_circ_0073128 with HOMER1B/C protein only in females. Global average area of Aβ deposits in entorhinal cortex samples was negatively correlated with *HOMER1B/C*, *HOMER1A* mRNA, and hsa_circ_0073127 in both genders. Furthermore, no differences in DNA methylation were found in two regions of *HOMER1* promoter between AD cases and controls. To sum up, we demonstrate that linear and circular RNA variants of *HOMER1* are downregulated in the entorhinal cortex of female patients with AD. These results add to the notion that *HOMER1* and its circular forms could be playing a female-specific role in the pathogenesis of AD.

## 1. Introduction

Alzheimer’s disease (AD) is a chronic, progressive and, for the time being, irreversible neurodegenerative process that is the leading cause of age-related dementia [1]. It is characterized by extracellular accumulation of amyloid plaques and intraneuronal tau neurofibrillary tangles causing loss of synapses and synaptic plasticity, thus impairing learning and memory [2,3,4,5]. It is further known that alterations in the homeostasis of glutamate and other ions such as Ca^2+^ are involved in the pathogenesis of AD [5,6].

Extracellular and intracellular β-Amyloid (Aβ) accumulation has been described in postmortem AD brains. Aβ peptide interferes with channels and receptors that regulate neuronal fundamental characteristics such as excitability and synapses [7]. In AD brains, it has been shown that aberrant excitability and synapses may be due to the interaction of Aβ with channels and receptors located at the synapse [7,8]. In this scenario, HOMER protein homolog 1 (*HOMER1*) gene plays a crucial role as a linker between neuronal channels and receptors. *HOMER1* belongs to HOMER family proteins which are involved in postsynaptic density (PSD) forming links between receptors, channels and other scaffolding proteins [9]. HOMER family consists of three members (*HOMER*1, 2, 3) with different isoforms each due to alternative splicing [10].

*HOMER1* has different messenger RNAs (mRNA) variants which can be grouped into long variants such as *HOMER1B* and *HOMER1C* and a short variant known as *HOMER1A* [11]. The main difference between them is the absence of exons 7 and 8 [12,13] and, in terms of protein expression, the absence of Coiled Coil domain (CC domain) in the short variant [8,14]. HOMER1 protein structure consists of an amino-terminal region containing 1–175 amino acids (EVH1 domain) and a proline-containing motif (Pmotif) [8,10]. Moreover, only long HOMER protein isoforms dimerize and bind through the CC domain to numerous components of postsynaptic density, such as Group1 metabotropic glutamate receptors (mGluR1), N-Methyl-D-aspartate (NMDA) receptor scaffolding, ankyrin repeat domains proteins (SHANK), and other receptors to regulate calcium signaling in excitatory synapses [8,11,15]. It is also well known that neural activity modulates gene expression by modifying alternative mRNA splicing, which in turn may change protein properties and therefore its function [14]. Hence, *HOMER1A* is encoded by an immediate-early gene (IEG), and its expression increases rapidly after neuronal activation [8]. In addition, the lack of the CC domain causes HOMER1A to act as a dominant-negative regulator of HOMER1B/C scaffolding capacity [11,15].

*HOMER1* mRNA isoforms are mainly expressed in the nervous system and play crucial roles in synaptic plasticity and signal transduction via their dynamic distribution at the postsynaptic level [8]. Changes in PSD formation induce alterations of the density and shape of the dendritic spines, impairing both learning and memory. Furthermore, the abnormal activation of glutamate receptor signaling and other cation channels are associated with a number of neurological diseases [8] such as mental retardation syndromes, schizophrenia, autism spectrum disorders or AD [8,9]. HOMER1B/C regulates the neuronal surface clustering of mGluRs, and treatment of fronto-cortical neurons with soluble Aβ results in rapid and significant thinning of the PSD and disassembly of HOMER1B/C synaptic clusters [8]. On another matter, *HOMER1A*, which is activity-dependent, has been suggested to regulate Aβ toxicity at the early stage of AD, and reduced *Homer1a* mRNA expression has been found in amyloid precursor protein and presenilin-1(APP + PS1) transgenic mice [8]. Yamamoto et al. [7] suggested that *HOMER1A* expression could be a therapeutic approach to prevent, at least partially, Aβ toxicity at the early stage of AD.

In addition to *HOMER1* linear RNA molecules, other RNA species have been described, such as *HOMER1* circular RNAs (circRNAs). circRNAs are a novel class of non-coding RNA characterized by a covalent junction between the 3′ end and 5′ end, formed by a back-splicing process. From the same pre-mRNA precursor, mRNAs and circRNAs are originated by canonical splicing and alternative splicing [16,17,18,19]. Though circRNAs are expressed all throughout the body, it is in the brain, and specifically at the synapses, where they are most expressed [20,21,22]. Some researchers have observed that a disproportionate fraction of circRNAs are derived from host genes that code for synaptic proteins [23]. In addition, it is known that circRNAs are playing important roles in brain development and in neurological diseases. In fact, altered expression in circRNAs in AD, and a *circHOMER1* transcript has been shown to be downregulated in different brain regions [24,25,26]. It is worth mentioning that this *circHOMER1* also is downregulated in prefrontal cortex and pluripotent stem-cell-derived neuronal cultures from patients with schizophrenia and bipolar disorder [11]. Thus, *circHOMER1* downregulation appears to be playing a role in synapses gene expression and cognitive flexibility [11].

Despite all these data, little is known about the expression of different *HOMER1* mRNA and circRNA transcripts in human AD brains. The purpose of this work is to provide knowledge about the expression of linear and circular *HOMER1* variants in the human AD entorhinal cortex, a brain region most vulnerable to the disease. We also seek to know whether epigenetic features, in particular DNA methylation levels, are altered in the promoter region of *HOMER1* in the brain of AD patients.

## 2. Results

To our knowledge, no previous studies have investigated *HOMER1* RNA variants expression in the human AD entorhinal cortex. As described above, HOMER1A protein acts as a negative regulator of HOMER1B/C, and both HOMER1A and HOMER1B/C play crucial roles in synaptic plasticity and signal transduction, processes that are altered in AD [8]. We therefore consider it interesting to study separately the expression of *HOMER1B/C*, which contains exons coding for the CC domain, and *HOMER1A*, which lacks these exons, in human entorhinal cortex affected by AD.

### 2.1. HOMER1 mRNA Variants Are Downregulated in Entorhinal Cortex of AD Female Cases Compared to Controls

Firstly, to test if *HOMER1* variants were differentially expressed in the entorhinal cortex of AD cases compared to controls, we measured *HOMER1* mRNA expression levels by RT-qPCR. All RNA samples passed the RNA quality threshold, so 28 AD cases were compared to 16 controls (Appendix A). To distinguish long from short *HOMER1* variant, specific primer sets were designed to be located at different exons (Figure 1A and Table 1). Following this approach, we found that *HOMER1B/C* mRNA levels showed a 0.61 fold decrease (*p*-value < 0.01) in AD cases compared to controls. *HOMER1A* mRNA expression levels were also reduced by 0.605 fold (*p*-value < 0.01) in AD cases compared to controls (Appendix A).

Next, we wanted to know whether these results were biased by age or gender, since we identified differences in age and gender between AD patients and controls (Appendix A). For this purpose, a univariate general linear model adjusted by age and gender was developed indicating that gender had influence on AD status (*p*-value = 0.003) while age did not (*p*-value = 0.486). To further assess any gender-related variation in *HOMER1* expression, the sample set was subdivided by gender and the analysis was redone in females (*n* = 23) and males (*n* = 21). Following this approach, we found females showed a significant decrease by 0.4 fold of *HOMER1B/C* and 0.55 fold of *HOMER1A* expression in AD samples compared to controls (*p*-value < 0.0001 and *p*-value < 0.01, respectively). However, no significant decrease was shown in male samples (*p*-value = 0.387 and *p*-value = 0.314, respectively) (Figure 1B).

We then sought to determine whether *HOMER1* mRNA variants expression was related to the severity of neuropathological changes of AD. When we analyzed expression according to the ABC score in females, *HOMER1B/C* mRNA showed significant differences, being its expression decreased in low (*p*-value < 0.01) and high (*p*-value < 0.05) ABC cases versus controls (Appendix A). On the contrary, no significant differences were found in the male subgroup. Regarding *HOMER1A* mRNA variant, no changes in expression levels were shown in either women or men.

Although ABC score is a very informative neuropathological score which includes Braak and Braak staging, we decided to analyze the severity also by Braak and Braak criteria, as it is still commonly used by investigators. In females, we found that *HOMER1B/C* mRNA expression levels were decreased in Braak I-II stage compared to controls (*p*-value < 0.05) and in Braak V-VI stage compared to controls (*p*-value < 0.05) (Appendix A), accordingly with ABC score results.

### 2.2. HOMER1 circRNA Variants Are Downregulated in Entorhinal Cortex in AD Female Cases Compared to Controls

Next, we wanted to test whether *HOMER1* circRNAs expression levels were also altered in AD entorhinal samples in AD females. For the purpose of this study, we selected three circRNAs from circInteractome [27], hsa_circ_0006916 and hsa_circ_0073128, which had been previously described in AD brains [24,26], and hsa_circ_0073127, which arises from exons 7 and 8 that are not included in the *HOMER1A* linear transcript. Thus, we designed specific primer sets to amplify and quantify each of these three *HOMER1* circRNAs (Figure 1A and Table 1) and Sanger sequencing was used to confirm the backsplice junction of the identified circRNAs.

We observed that hsa_circ_0006916 and hsa_circ_0073127 expression levels were downregulated in the entorhinal cortex of female AD cases compared to female controls (fold-change = 0.39, *p* < 0.05 and fold-change = 0.46, *p* < 0.01, respectively) (Figure 1C). By contrast, hsa_circ_0073128 expression level did not change, neither change any *circHOMER1s* in the male subgroup.

We also analyzed *circHOMER*1s expression according to ABC score and gender subgroup. In females, the three *circHOMER1*s expression levels showed significant differences, being its expression decreased in low ABC cases versus controls (hsa_circ_006916, *p*-value < 0.01; hsa_circ_0073127, *p*-value < 0.05; and hsa_circ_0073128, *p*-value < 0.05) (Appendix A). No differences in *HOMER1* circRNAs according to ABC score were found in the male subgroup. When considering only Braak staging (Appendix A), no differences were observed for any *circHOMER*1 in either women or men.

### 2.3. Proportional Decreases in HOMER1 mRNA and circRNAs Variants Expression Levels in AD Female Cases Compared to Controls

To answer the question of whether or not the change in expression levels between controls and AD female cases is proportional across the different mRNA *HOMER1* variants and circRNAs, we used a univariate general linear model. In this model the dependent variable was the expression of the different *HOMER1* RNA transcripts and the predictors were the specific RNA variants and disease status (AD or control).

Overall, expression of all variants together was 56.43% lower in AD females compared to control cases. The variant most highly expressed was hsa_circ_0006916, even more highly expressed than linear *HOMER1* variants (Appendix A). Furthermore, expression levels of variants were different among them. Considering *HOMER1A* as a reference, expression of *HOMER1B/C* was 70.81% lower, and hsa_circ_0073127 and hsa_circ_0073128 were expressed 93.81% and 18.92% less, respectively. On the contrary, hsa_circ_0006916 showed 440.01% higher expression compared to *HOMER1A*. However, no significant differences were shown in expression when we consider cases and each transcript together in the model, meaning that the decrease in expression of the different variants is proportional within AD female cases (Appendix A). We conclude that relative expression ratios of each transcript are maintained even though they are significantly decreased in AD females compared to controls when studied independently.

### 2.4. Correlation between HOMER1 Protein and RNA Expression Levels in the Entorhinal Cortex of Females

In order to test if downregulation of *HOMER1* mRNA in AD cases was related with protein levels, Western blot analysis was carried out in 15 entorhinal human samples: 6 controls (3 females) and 9 AD cases (6 females). We observed that HOMER1 B/C protein levels were downregulated in AD brain cases compared to controls (fold-change = 2.19, *p*-value < 0.01) (Appendix A). However, when we re-performed the analysis by gender subgrouping neither female or male showed significant differences between controls and AD samples.

Next, we analyzed the correlation between *HOMER1* RNA variants expression levels and protein expression levels by gender subgroup. We observed in females a positive correlation between mRNA forms and protein expression of *HOMER1B/C* (r = 0.733, *p*-value < 0.05) and a strong positive correlation between hsa_circ_0073128 and protein expression (r = 0.883, *p*-value < 0.01). However, we found no correlation between protein and any RNA expression levels in males.

### 2.5. Correlation between DNA Methylation Levels in HOMER1 Promoter and RNA Expression Levels

DNA methylation levels in regulatory regions of the genome are known to modulate the expression of related or nearby genes. To determine whether DNA methylation within the *HOMER1* promoter region is related with *HOMER1* expression and altered in the brain of AD patients, we explored two regions within the promoter of *HOMER1* gene, one of them overlapping a CpG island. To this end, we used bisulfite sequencing cloning (Figure 2A) in genomic DNA isolated from entorhinal cortex of four AD cases and four controls.

These regions showed low methylation levels (Figure 2B) as corresponding to genes actively expressed. We observed a positive correlation between DNA methylation levels of the promoter region overlapping the CpG island and mRNA expression levels of *HOMER1A* (r = 0.889, *p*-value < 0.001) and a statistical trend was found for *HOMER1 B/C* (r = 0.741, *p*-value = 0.057). On the other hand, no significant correlation was shown between *HOMER1* circRNA variants and DNA methylation levels. In addition, no differences in DNA methylation were observed in either region between AD cases and controls.

### 2.6. Correlation of HOMER1 RNA Variants Expression Levels with Aβ Deposits

We next aimed to correlate *HOMER1* RNA variants expression levels with the burden of Aβ in the AD brains. In the female group the global average area of Aβ deposits in entorhinal cortex was negatively correlated with linear *HOMER1* RNA variants levels (Table 2) and hsa_circ_0073127. Males also showed negative correlation between the global average area of Aβ deposits in entorhinal cortex and linear *HOMER1B/C* and circRNAs.

## 3. Discussion

In this study, we show that linear and specific circular variants of *HOMER1* RNA are downregulated in the human entorhinal cortex of AD female patients while no changes are found in males. In addition, a positive correlation of *HOMER1* linear RNAs and a *circHOMER1* with protein expression was observed. Moreover, there is a negative correlation of *HOMER1* RNA variants expression levels with Aβ burden in both genders at human entorhinal cortex. Our data also suggest that DNA methylation at the promoter region of *HOMER1* is not altered in AD and therefore is probably not involved in regulating the lower expression of *HOMER1* RNA observed in the disease.

The *HOMER1* gene is expressed in the PSD mainly at glutamatergic synapses [15]. Due to alternative splicing two isoforms have been described; *HOMER1B/C* plays a role in synapses as a scaffolding protein, establishing connections between receptors involved, among other processes, in Ca^2+^ homeostasis [15]. Regarding *HOMER1A*, it is a negative regulator of *HOMER1B/C* scaffolding capacity. *HOMER1* is considered an IEG, as it is induced in response to neuronal activity [9] and is a crucial gene for synapses plasticity and memory consolidation [8,28].

The results of the present study are consistent with previous reports where deregulation of *HOMER*1 was described in brain tissue from other neurological conditions [8,9,29]. *HOMER1A* expression is downregulated in some particular brain areas in bipolar disorder, while in others is upregulated [29]. In schizophrenia, the *HOMER1A/HOMER1B/C* ratio was observed to be altered in hippocampus, being *HOMER1B/C* downregulated and *HOMER1A* upregulated [30]. In addition, brain ischemia seems to downregulate expression levels of *HOMER1A* and *HOMER B/C* [31]. However, to our knowledge, no previous report of gender-specific *HOMER1* expression in brain tissue have been reported.

Since *HOMER1* is a major component of synapses, it is not surprising that this gene is found altered in synaptopathies, including some of the above-mentioned disorders. Indeed, many neurological and psychiatric conditions show dysfunctional synapses. In the case of AD, synapses are thought to be altered due to the toxic effect of amyloid peptides and tau protein tangles [32]. In fact, previous reports have suggested that *Homer1a* could be involved in Aβ toxicity at the early stage of AD [8]. Even a role in processing Aβ has been proposed for *Homer1a* based on studies performed in other animal models [33]. In our study, we observed in both genders a negative correlation between the burden of Aβ and expression levels of all the linear transcripts and some circRNAs. On the whole, these data suggest a relationship between the *HOMER1* gene and Aβ deposition and support the idea that HOMER1 might be involved in Aβ processing, as other authors have proposed [33]. Nevertheless, the observation of reduced expression levels of *HOMER1* RNA transcripts in brain tissue may well be a consequence of reduced synapsis typical of AD and not the other way around. To elucidate whether Aβ or tau have an effect on neuronal *HOMER1* expression, or if *HOMER1* definitely influences Aβ deposition, in vitro or in vivo experiments are needed, and the present study only shows an association between both variables.

The present study shows an intriguing result, being *HOMER1* downregulated only in the brain of female patients with AD. This precludes extending the conclusions to men with AD. However, it is well known that a gender bias has long been observed in AD, with women being at the highest risk of developing the disease [34,35]. Nevertheless, the molecular basis of this gender bias is still largely unknown. Interestingly, estradiol seems to prevent hippocampal neuronal death with exposure to glutamate [36]. Additionally, some authors have observed a female-selective reduction in GluR2 AMPA glutamate receptor subunit expression in the nucleus basalis neurons in AD, suggesting that cholinergic nucleus basalis neurons may be at greater risk for degeneration during the progression of the disease in females [37]. The cause of a female bias in AD is still intriguing and brain molecular changes underlying AD pathology may help to better understand this epidemiological bias. Indeed, to focus on gender differences in AD has been strongly recommended by experts in the field [38]. In this line, sex-specific genetic variants in AD biomarkers have been also reported [39].

*HOMER1A* has been involved in glutamate-dependent synaptic plasticity [40]. Increasing evidence points to glutamate-mediated excitotoxicity in development of neuronal death in AD and other neurodegenerative conditions [41]. A prolonged increase in glutamate over time triggers a postsynaptic response, and thus, an increase in Ca^2+^ concentration in neurons being neurotoxic [5]. Importantly, elucidation of the molecular bases underlying synaptopathies may help to discovered new drug targets for these disorders, including AD. In this scenario, *Homer1a* has been shown to be upregulated by memantine [40], which is a non-competitive NMDA receptor antagonist used to alleviate clinical symptoms of AD [40]. Accordingly, *Homer1* is also altered in the hippocampus of a murine model used to investigate the effectiveness of acetylcholinesterase inhibitors in the treatment of AD [42]. As we observed a decrease in the long isoform, *HOMER1B/C*, at early stages of AD, drugs that increase *HOMER1A* or *HOMER1B/C* brain expression would be interesting to be explored in AD female patients.

Most interestingly, a relationship between sleep deprivation and Aβ brain burden has been recently recognized [43,44,45,46]. Sleep disturbance is related to cognitive impairment and may be a risk factor for AD by increasing Aβ deposition [45]. *HOMER1* arises as a candidate to explain, at least in part, the link between sleep alterations and AD dementia, since *HOMER1* is a gene closely related to sleep and a crucial marker of sleep deficit. For instance, it is well known that *Homer1a* expression undergoes circadian regulation in the brain [47,48] and Homer1a is thought to mediate the cross-talk between circadian inputs and neuronal activity. Recently, it has been described that *HOMER1* mRNA expression is high at baseline in human brain regions where Aβ accumulates when sleep is altered [49].

One of the roles of circRNAs is to regulate linear RNA transcription (including its host RNA) but also other epigenetics mechanisms could be regulating mRNA and circRNA expression. In cancer, CpG promoter hypermethylation has been shown to downregulate circRNA and its host linear RNA expression [50]. Interestingly, it has been described that in fear conditions or depression there exists an epigenetic modulation of *HOMER1* transcription regulation [51,52]. It is also worth mentioning that changes in epigenetic marks have been observed in human AD brains [53,54]. Alterations in CpG methylation could be playing a role in the different circular and linear *HOMER1* variants expression in AD. We explored the status of DNA methylation at the promoter region of *HOMER1* and found a positive relationship between methylation levels and mRNA expression of *HOMER1A* variant. However, we did not find significant correlation with HOMER1 B/C or circRNA expression levels or any significant change between patients and controls. This may suggest that expression of *HOMER1* linear and circular transcripts are not regulated on the whole by DNA methylation. Nevertheless, we should be very cautious with this idea since the sample size is small and only two amplicons have been surveyed, and methylation at other regions such as enhancers or alternative promoters could be playing a part in modulating *HOMER1* expression. To assess whether DNA methylation within *HOMER1* promoter is involved in downregulation of *HOMER1* transcripts in AD cases, further experiments would be required.

It is interesting that we also found a consistent decreased expression of two circRNAs in the entorhinal cortex of AD female patients, i.e., hsa_circ_0006916 and hsa_circ_0073127; hsa_circ_0006916 was first described as a circRNA derived from the synaptic *HOMER1* gene [23]. Our result is concordant with a previous study where a RNAseq experiment was performed in human parietal lobe tissue to identify and quantify circRNAs related to AD [24]. Interestingly, hsa_circ_006916 was also downregulated in the parietal and frontal cortex of AD patients [24,25]. A more recent study also found decreased expression levels of hsa_circ_006916 in the anterior prefrontal cortex, inferior frontal gyrus and parahippocampal gyrus, and of hsa_circ_0073128 in the anterior prefrontal cortex [26]. Intriguingly, in the latter study hsa_circ_006916 showed no differential changes in AD in the superior temporal lobe and neither did hsa_circ_0073128 in the superior temporal lobe, inferior frontal gyrus and parahippocampal gyrus. In our study, hsa_circ_0073127 arises as a novel circRNA consistently involved in AD, at least in females, since it was downregulated in AD female cases. Moreover, we want to highlight that hsa_circ_006916 was the majoritarian variant in *HOMER1* expression levels, with significantly higher levels than linear *HOMER1* RNA variants. Other genes have been previously described to show a predominant expression of a circRNA variant compared to its mRNA cognates, including the AD-related *CDR1* gene [20,23,24]. It is very relevant that circRNAs seem to be more enriched in synaptic processes than their linear isoforms, and *HOMER1*, as a synaptic gene, seems to follow this rule [23]. The results of our study and others point to a coordinated *HOMER1* circRNA downregulation across different brain regions. Our results in the entorhinal cortex, a brain area where neuropathological changes of AD start, may also suggest an early involvement of *HOMER1* changes, including its circRNA variants, in the development of the disease in females. However, we should be very cautious, since robustness of the stratified analysis may be limited by sample size, and this is only an observational study that precludes drawing conclusions about mechanistic relationships. Further studies on *HOMER1* expression in other brain series would add evidence to definitively unravel the real involvement of the *HOMER1* gene in AD pathology.

In conclusion, specific *HOMER1* circular and linear RNA transcripts are decreased in the entorhinal cortex of AD patients, suggesting that this synapse-related gene may play a crucial role in the early development of the disease. Moreover, we describe a novel circRNA, hsa_circ_0073127, associated with AD pathology in a gender-specific manner. Gender-specific analysis may improve understanding of AD pathogenesis and reinforce the use of personalized medicine. As is the case of *HOMER1,* circRNA species of specific genes seem to be associated with amyloid deposits, which opens the path for studying these molecules as biomarkers of synaptopathies or even novel pharmacological targets for AD.

## 4. Materials and Methods

### 4.1. Human Entorhinal Samples

Brain entorhinal cortex samples from 28 AD patients and 16 controls were provided by Navarrabiomed Brain Bank. After death, half brain specimens from donors were cryopreserved at −80 °C. Neuropathological examination was completed following the usual recommendations [55] and according to the updated National Institute on Aging-Alzheimer’s Association guidelines [56].

Assessment of Aβ deposition was carried out by immunohistochemical staining of paraffin-embedded sections (3–5 μm thick) with a mouse monoclonal (S6 F/3D) anti-Aβ antibody (Leica Biosystems Newcastle Ltd., Newcastle upon Tyne, United Kingdom). Evaluation of neurofibrillary pathology was performed with a mouse monoclonal antibody anti-human PHF-TAU, clone AT-8 (Tau AT8) (Innogenetics, Gent, Belgium), which identifies hyperphosphorylated tau (*p*-tau) [57]. The reaction product was visualized using an automated slide immunostainer (Leica Bond Max) with Bond Polymer Refine Detection (Leica Biosystems, Newcastle Ltd., Newcastle upon Tyne, United Kingdom). Other protein deposits, such as synuclein deposits, were ruled out by a monoclonal antibody against α-synuclein (NCL-L-ASYN; Leica Biosystems, Wetzlar, Germany). The staging of AD was performed by using the ABC score according to the updated National Institute on Aging-Alzheimer’s Association guidelines [56]. ABC score combines histopathologic assessments of Aβ deposits determined by the method of Thal (A) [56], staging of neurofibrillary tangles by Braak and Braak classification (B) [57], and scoring of neuritic plaques by the method of CERAD (Consortium to Establish A Registry for Alzheimer’s Disease) (C) [58] to characterize AD neuropathological changes. Thus, the ABC score shows three levels of AD neuropathological severity: low, intermediate, and high. A summary of the characteristics of subjects considered in this study is shown in Appendix A.

### 4.2. Quantitative Assessment of Aβ Deposits in Brain Tissues

In order to quantitatively assess the Aβ burden for further statistical analysis, we applied a method to quantify protein deposits. This method generates a numeric measurement that represents the extent of Aβ deposition. Sections of the entorhinal cortex were examined after performing immunostaining with anti-Aβ antibody as described above in human entorhinal samples. Three pictures were obtained for each immunostained section by using an Olympus BX51 microscope at ×10 magnification power. Focal deposit of Aβ, as described by Braak and Braak (neuritic, immature, and compact plaque) [57], was manually determined and was further edited and analyzed with the ImageJ software. Then, Aβ plaque count, referred to as amyloid plaque score (APS) and total area of Aβ deposition were automatically measured by ImageJ and averaged for each section.

### 4.3. RNA Isolation and RT-qPCR

Total RNA was isolated from entorhinal cortex with RNAeasy Lipid Tissue mini-Kit (QIAGEN, Redwood City, CA, USA) following the manufacturer’s instructions. Genomic DNA was removed with recombinant DNase (TURBO DNA-free™ Kit, Ambion Inc., Austin, TX, USA). Concentration and purity of RNA were both evaluated with NanoDrop spectrophotometer. Complementary DNA (cDNA) was reverse transcribed from 500 ng total RNA with SuperScript^®^ III First-Strand Synthesis Reverse Transcriptase (Invitrogen, Carlsbad, CA, USA) after priming with random primers. RT-qPCR reactions were performed in triplicate with Power SYBR Green PCR Master Mix (Invitrogen, Carlsbad, CA, USA) in a QuantStudio 12K Flex Real-Time PCR System (Applied Biosystems, Foster City, CA, USA) and repeated twice within independent cDNA sets. Sequences of primer pairs were designed using Real Time PCR tool (IDT, Coralville, IA, USA) and Primer3 software and are listed in Table 1. Relative expression level of mRNA in a particular sample was calculated as previously described [59] and geometric mean of *GAPDH* and *ACTB* genes were used as the reference to normalize expression values.

### 4.4. circRNA Confirmation by Sanger Sequencing

circRNA candidate bands were selected after 1.8% agarose gel electrophoresis of RT-qPCR products. Bands purification was made with Wizard^®^ SV Gel and PCR Clean-Up System (Promega, Madison, WI, USA). Next, Sanger sequencing was performed and UCSC Genome Browser software Blat-tool was used for sequence alignment [60,61].

### 4.5. HOMER1 Protein Expression Analysis by Western Blot

Human entorhinal cortex tissue from patients and control samples was lysed with 100 μL lysis buffer containing urea, thiourea, and DTT. After centrifugation at 35,000 rpm for 1 h at 15 °C, extracted proteins were quantified following the Bradford Protein Assay (Bio-Rad, Hercules, CA, USA) by using a spectrophotometer. Next, 10 μg of protein per sample were resolved in 4–15% Criterion TGX stain-free gels (Bio-Rad, Hercules, CA, USA) and electrophoretically transferred onto nitrocellulose membranes using a Trans-blot Turbo transfer system (25 V, 7 min) (Bio-Rad, Hercules, CA, USA). Equal loading of the gel was assessed by stain-free digitalization and by Ponceau staining. Membranes were probed with mouse anti-human HOMER1 B/C primary antibody (Santa Cruz sc-25271; 1:250) (Santa Cruz Biotechnology Inc., Dallas, TX, USA) in 5% nonfat milk and incubated with peroxidase-conjugated anti-mouse secondary antibody (Bio-Rad, 170-6516, 1:2000) (Bio-Rad, Hercules, CA, USA). Immunoblots were then visualized by exposure to an enhanced chemiluminescence Clarity Western ECL Substrate (Bio-Rad, Hercules, CA, USA) using a ChemidocMP Imaging System (Bio-Rad, Hercules, CA, USA). Expression levels of HOMER1 B/C were standardized by the corresponding band intensity of mouse anti-human GAPDH (Calbiochem; cb1001, 1:10,000).

### 4.6. HOMER1 Methylation by Bisulfite Cloning Sequencing

Bisulfite-converted genomic DNA was used to study promotor methylation levels of HOMER1 gene. Primer pair sequences were designed by MethPrimer [62] and are listed in Table 1. PCR products were cloned using the TopoTA Cloning System (Invitrogen, Carlsbad, CA, USA), and a minimum of 10–12 independent clones were sequenced for each examined subject and region. Methylation graphs were obtained by using QUMA software [63].

### 4.7. Statistical Data Analysis

Statistical analysis was performed with SPSS 25.0 (IBM Corp., Armonk, NY, USA). Before performing differential analysis, we checked whether continuous variables were normally distributed, as per one-sample Kolgomorov–Smirnov test and the normal quantil–quantil (QQ) plots. Statistical significance was set at *p*-value < 0.05. Mann–Whitney *U* test was used to assess statistical difference of the following variables: mRNA, circRNAs and protein levels.

A univariate general linear model adjusted by age and gender was used to know whether the difference of the mRNA and circRNAs expression levels were biased by age or gender, the values of expression levels were transformed by using log_10_(X) to meet the conditions of normality. Kruskal–Wallis one-way analysis of variance (ANOVA) was used to analyze differences in the expression levels of *HOMER1* mRNA and circRNAs variants among ABC stage groups and Braak stage. Univariate general linear model was used to study the proportions between *HOMER1* RNA variants expression levels in AD versus control samples. Previously, the values of expression levels were transformed by using log_10_(X) to meet the conditions of normality. Differences in methylation levels between two bisulfite cloning sequencing groups was evaluated with Mann–Whitney *U* test. Spearman test was used to assess correlation between continuous variables. GraphPad Prism version 6.00 for Windows (GraphPad Software, La Jolla, CA, USA) was used to draw graphs except for methylation figures that were drawn by QUMA software.

## Figures and Tables

**Figure 1 ijms-22-09205-f001:**
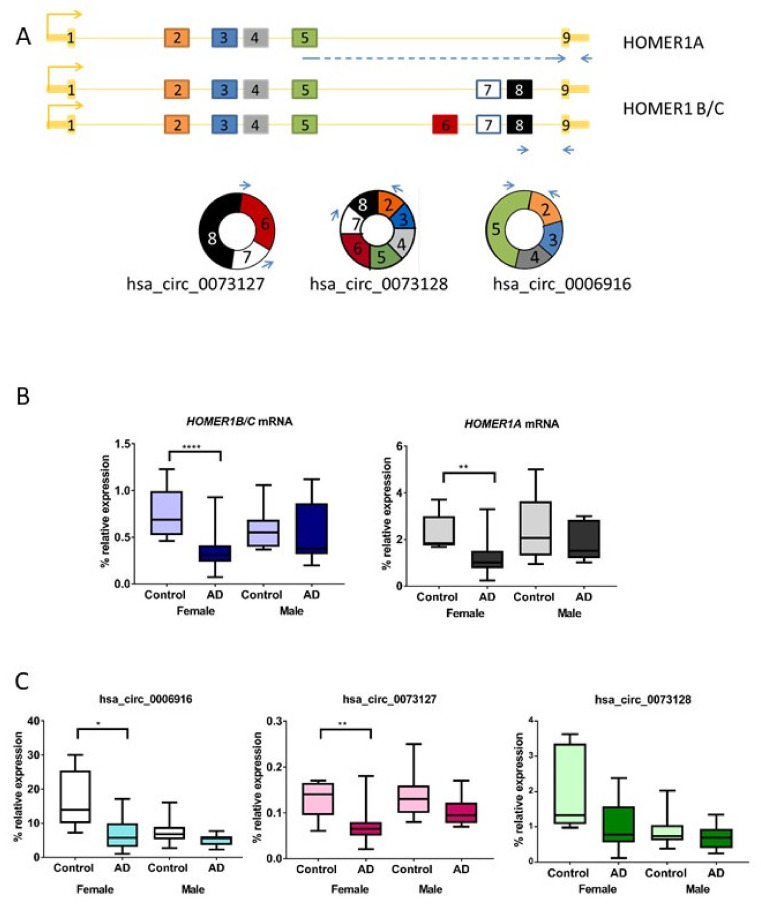
*HOMER1* and *circHOMER1* variants expression. (**A**) Gen map for each linear and circular *HOMER1* RNA form. Boxes represent exons and arrows show where primers are located. (**B**,**C**) Box-plot represents the percentage of *HOMER1* and *circHOMER1* expression relative to geometric mean of *GAPDH* and *ACTB* housekeeping genes expression in females (*n* = 23) and males (*n* = 21). * *p*-value< 0.05; ** *p*-value < 0.01; **** *-*value < 0.0001.

**Figure 2 ijms-22-09205-f002:**
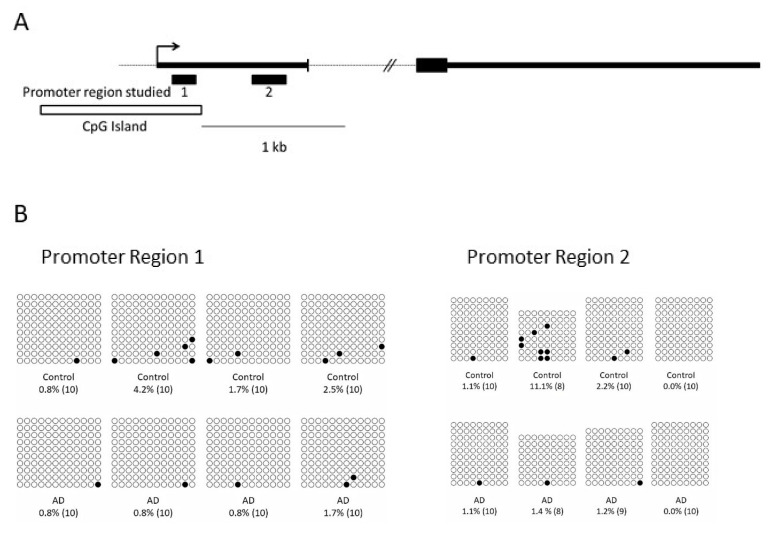
DNA methylation levels in entorhinal samples. (**A**) The graph shows genomic position of the amplicons (black boxes) assessed by bisulfite cloning sequencing within the exon 1 of *HOMER1* gene. CpG island is represented by white box. (**B**) Bisulfite cloning sequencing for the two amplicons studied. Black and white circles denote methylated and unmethylated cytosines, respectively. Each column symbolizes a unique CpG site in the examined amplicon, and each line represents an individual DNA clone.

**Table 1 ijms-22-09205-t001:** Primers sets.

	Forward 5’->3’	Reverse 5’->3’
*HOMER1A*	CACATAGGAAGTAGAAATTCGGAAC	TTTCACTTTCCTTAGCTGCATTC
*HOMER1B/C*	CAGAGAACTACAAGAACAGAGGG	GACGTTGCTCTAAGTCAGACAG
hsa_circ_0006916	TTTGGAAGACATGAGCTCGA	AAGGGCTGAACCAACTCAGA
hsa_circ_0073127	GAAACTACAGTTCAGCAATCAGC	AAGTTCAGTCACCCGCTTGT
hsa_circ_0073128	AGCCAAGCAAATGCAGTACA	TGTGTTTGGGTCAATTTGGA
Bis-*HOMER1_1*	GATTTTATTTTTTTTAGTTTTTTTT	TAACTCTATAATTCATTCATTCTCC
Bis-*HOMER1*_2	TTTTTTAGGTAAAATGATTTTTTTT	AATTACCTAACTCTCCCTAAATATC

**Table 2 ijms-22-09205-t002:** Correlation between global average area of Aβ deposits and *HOMER1* linear RNAs and circRNAs expression levels. (*n* = 20). * *p*-value < 0.05.

Female		*HOMER1B/C*	*HOMER1A*	hsa_circ_0073127	hsa_circ_0073128	hsa_circ_0006916
global average area of Aβ deposits	correlation coeficient (Spearman’s Rho)	−0.483 *	−0.447 *	−0.479 *	−0.005	−0.038
	Sig. (bilateral)	0.036	0.048	0.038	0.982	0.873
Male		*HOMER1B/C*	*HOMER1A*	hsa_circ_0073127	hsa_circ_0073128	hsa_circ_0006916
global average area of Aβ deposits	correlation coeficient (Spearman’s Rho)	−0.487 *	−0.452	−0.494 *	−0.473 *	−0.486 *
	Sig. (bilateral)	0.041	0.059	0.037	0.047	0.041

## Data Availability

Not applicable.

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
