# Peer review of "Gender-Dependent Deregulation of Linear and Circular RNA Variants of HOMER1 in the Entorhinal Cortex of Alzheimer’s Disease"

_ijms, 2021, doi:10.3390/ijms22179205_

Round 1

Reviewer 1 Report

The authors study the expression of different isoforms of HOMER1 gene in control and Alzheimer´s disease patients. Some of these isoforms of HOMER changes their protein level in the entorhinal cortex of Alzheimer´s disease patients.

The conclusions are not convincing since there is a lot of difference between the control and the AD group. Meaning that in the control group around 42% of the cases are over 60 years-old adults and around 37% are females while in the AD group around 96% are over 60 years old and 64% are females. It is well known that the protein levels change with age and there are gender differences in AD. An explanation/comment of this is missing in the discussion.

-Prior studies are referenced apropriately and the whole text is easy to follow. Text and figures are quite clear. However, I suggested to add a bit of color to the figures, so it is easier to read the graphs. For example: use a different color for each study group. As well, clarify which controls and AD are used in each analysis. For example a) in figure 1: which cases have been used for the westerblot?; b) in figure S1: in the “control” and “AD” groups, which case has been used in figure C and D?
-The study is interesting but in order to be convincing it is needed to include more analysis. It is missing: gender and age analysis; relationship between HOMER levels and Braak stage; relationship between HOMER levels and % b-amiloid plaque area.

Author Response

Point 1: The authors study the expression of different isoforms of HOMER1 gene in control and Alzheimer´s disease patients. Some of these isoforms of HOMER changes their protein level in the entorhinal cortex of Alzheimer´s disease patients.

The conclusions are not convincing since there is a lot of difference between the control and the AD group. Meaning that in the control group around 42% of the cases are over 60 years-old adults and around 37% are females while in the AD group around 96% are over 60 years old and 64% are females. It is well known that the protein levels change with age and there are gender differences in AD. An explanation/comment of this is missing in the discussion.

Response 1: We thank the Reviewer#1 for this constructive comment that has helped to definitely improve the manuscript. Our sample set is indeed heterogeneous in age and gender.

The differences are due, in part, to the fact that the samples selected for this study were samples with pure Alzheimer's disease, i.e., without other protein deposits, and the controls were also pure, without any protein deposits. This condition has limited the availability of samples in the biobank, since it is very difficult to have control samples at late ages without any protein deposits.

Consequently, and in the light of the reviewers' comments, we have decided to reanalyze the results taking into account the differences in age and gender of the sample.

Thus, we have developed a univariate general linear model adjusted by age and gender and observed that age does not influence the differences in HOMER1 expression between AD patients and controls while gender does. We then decided to subdivide the sample by gender and reanalyze and the results were certainly very interesting. The downregulation observed in AD entorhinal cortex is mainly driven by a downregulation in the female group. Thus, the results section has been rewritten based on this analysis and the discussion has been updated accordingly. The title has also been changed to address the gender-specific nature of the results.

Point 2: -Prior studies are referenced appropriately and the whole text is easy to follow. Text and figures are quite clear. However, I suggested to add a bit of color to the figures, so it is easier to read the graphs. For example: use a different color for each study group.

Response 2: Following the reviewer advice, we have added some color to the figures which have definitively improve the clarity of the charts.

Point 3: As well, clarify which controls and AD are used in each analysis. For example a) in figure 1: which cases have been used for the westerblot?

Response 3: Western blot is now shown in a supplemental figure 4. For WB we have used 15 entorhinal human samples: 6 controls (3 females) and 9 AD cases (6 females.)

Point 4: b) in figure S1: in the “control” and “AD” groups, which case has been used in figure C and D?

Response 4: We have used all samples which are included in supplementary table 1.

Now, these figures are in supplemental figure 3 A,B and we have used all female samples

Point 5: -The study is interesting but in order to be convincing it is needed to include more analysis. It is missing: gender and age analysis; relationship between HOMER levels and Braak stage; relationship between HOMER levels and % b-amiloid plaque area.

Response 5: As mentioned in the first comment, we have conducted further analysis to improve the article. In particular, we have performed an analysis by gender, showing that the differences in HOMER1 expression levels between AD patients and controls are maintained only in the group of women.

In relation to the analysis by Braak & Braak, we would like to mention that the ABC score is a more comprehensive neuropathological classification than that of Braak & Braak, which also includes it, the latter being the second item that integrates the ABC classification (B is for Braak & Braak). In spite of this, and since Braak & Braak is a classification still widely used by researchers, we have performed an additional analysis, which shows similar results to those obtained when stratifying by ABC score.

Finally, the analysis between HOMER1 expression and beta amyloid plaque area percentage was already included in our first version of the article (results section 2.6).

Reviewer 2 Report

In this Manuscript, the Authors show their study about HOMER1 RNA forms in entorhinal cortex from human frozen brain samples.  They compare AD and controls finding a general downregulation of this synaptic gene, also at protein level. They speculate an association with sleep deisorder and amyloid burden.

In my view, the main limitation of the presented research is the lack of matching between AD cases and controls, particularly for age (controls are significantly younger basing on Table 1). This introduces a possible bias for their analysis and conclusions. I invite the Authors to consider this and try to run their analysis in a more appropriate control group selected from the already used one. If this is not possible, results should be tuned down accordingly and this aspect properly commented in the Discussion section as main limitation. Of course, a real improvement would be adding more age-matched controls-

A similar critique may apply to the statification according to ABC criteria and disease severity. Numbers are quite small and stratification robustness limited.

Minor points

1) Figure 1. The Western blot  of HOMER shows two bands, with variable intensity also among cases or AD. Please explain which band was considered specific or provide more details on this pattern.

2) The negative correlation between amyloid burden and HOMER level of expression may be a consequence of synaptic failure in AD brains and not a mechanistic correlation. This should be more properly discussed. Moreover, it is true that this is one of the firstly involved areas in AD, but I think that a speculation for an early role in pathogenesis is to be tuned down, as well as the possible link with sleep, unless the Authors are able to provide detailed clinical information for the included AD cases and their sleep disturbances.  

Author Response

Point 1:In this Manuscript, the Authors show their study about HOMER1 RNA forms in entorhinal cortex from human frozen brain samples.  They compare AD and controls finding a general downregulation of this synaptic gene, also at protein level. They speculate an association with sleep disorder and amyloid burden.

In my view, the main limitation of the presented research is the lack of matching between AD cases and controls, particularly for age (controls are significantly younger basing on Table 1). This introduces a possible bias for their analysis and conclusions. I invite the Authors to consider this and try to run their analysis in a more appropriate control group selected from the already used one. If this is not possible, results should be tuned down accordingly and this aspect properly commented in the Discussion section as main limitation. Of course, a real improvement would be adding more age-matched controls-

A similar critique may apply to the stratification according to ABC criteria and disease severity. Numbers are quite small and stratification robustness limited.

Response 1: We sincerely thank reviewer #1 for the interesting suggestion of running the analysis in a more appropriate manner. Unfortunately, additional age-matched controls were not available. The design of our study focused on identifying HOMER1 RNA expression levels in “pure” AD cases, that is to say, in patients with no other protein deposits than amyloid and tau to avoid spurious findings due to the presence of other molecular disturbances. In the case of controls, we were also very stringent and only brains with none protein deposit were selected. This approach maximizes chances of finding true molecular associations with AD, even though reducing the number of older “pure” controls, which are definitely scarce in the biobanks. As a consequence, our controls were younger than ADs.

In the new version of the manuscript, we developed a univariate general linear model adjusted by age and gender and observed that age does not influence the differences in HOMER1 expression between AD patients and controls while gender does. We then decided to subdivide the sample by gender and reanalyze and the results were certainly very interesting. The downregulation observed in AD entorhinal cortex is mainly driven by a downregulation in the female group. Thus, the results section has been rewritten based on this analysis and the discussion has been updated accordingly.

Point 2: Minor points

Figure 1. The Western blot  of HOMER shows two bands, with variable intensity also among cases or AD. Please explain which band was considered specific or provide more details on this pattern.

Response 2: We have taken both bands into account, since two bands appear in the figure of the antibody data sheet in the mouse whole brain lysate. This has been clarified in the legend of the WB figure (Supplemental Figure 4).

Point 3: The negative correlation between amyloid burden and HOMER level of expression may be a consequence of synaptic failure in AD brains and not a mechanistic correlation. This should be more properly discussed. Moreover, it is true that this is one of the firstly involved areas in AD, but I think that a speculation for an early role in pathogenesis is to be tuned down, as well as the possible link with sleep, unless the Authors are able to provide detailed clinical information for the included AD cases and their sleep disturbances.

Response 3: We agree with the reviewer that negative correlation between amyloid burden and HOMER1 RNA expression levels may well be a consequence of synaptic failure and not the other way around. Indeed, this is an observational study and no cause-consequence conclusions cannot be drawn. Only new hypothesis may arise from the conclusions and further experimental studies are required to assess this point. Thus, we have acknowledged this fact in the discussion section (lines 283-285).

We have also moderated the conclusions and comments in regard to the early stages of the disease and the possible association with sleep. The following text has been deleted:

“In our study, HOMER1 downregulation is significant mainly at early stages of AD neuropathological changes which is in line with the idea that sleep disturbances are involved in the pathogenesis of AD particularly at early stages of neurodegeneration”

“HOMER1 is a crucial marker of sleep deprivation, these results add evidence to the bi-directional relationship of AD and sleep disturbances.”

And the following text has been added to the Discussion section:

“However, we should be very cautious since robustness of the stratified analysis may be limited by sample size and this is only an observational study that precludes drawing conclusions about mechanistic relationships. Further studies on HOMER1 expression in other brain series would add evidence to definitively unravel the real involvement of the HOMER1 gene in AD pathology.

Round 2

Reviewer 1 Report

The authors have addressed the major concerns raised by this reviewer and there is no additional comments. 

Reviewer 2 Report

In this revised form, the Authors have tried to improve their data analysis and have better stressed limitations and speculations that are there.

I have no additional coments, thanks.